# Effect of Erythropoietin on the Expression of Murine Transferrin Receptor 2

**DOI:** 10.3390/ijms22158209

**Published:** 2021-07-30

**Authors:** Betty Berezovsky, Martin Báječný, Jana Frýdlová, Iuliia Gurieva, Daniel Wayne Rogalsky, Petr Přikryl, Vít Pospíšil, Emanuel Nečas, Martin Vokurka, Jan Krijt

**Affiliations:** Institute of Pathophysiology, First Faculty of Medicine, Charles University, U Nemocnice 5, 128 53 Prague, Czech Republic; bettyberezovsky@gmail.com (B.B.); mbaje@lf1.cuni.cz (M.B.); jana.frydlova@lf1.cuni.cz (J.F.); gurieva.jul@seznam.cz (I.G.); danielrogalsky12@gmail.com (D.W.R.); pprik@lf1.cuni.cz (P.P.); vitek_pos@hotmail.com (V.P.); necas@cesnet.cz (E.N.); mvoku@lf1.cuni.cz (M.V.)

**Keywords:** hepcidin, erythroferrone, transferrin receptor, exosomes

## Abstract

Erythropoietin (EPO) downregulates hepcidin expression to increase the availability of iron; the downregulation of hepcidin is mediated by erythroferrone (ERFE) secreted by erythroblasts. Erythroblasts also express transferrin receptor 2 (TFR2); however, the possible role of TFR2 in hepcidin downregulation is unclear. The purpose of the study was to correlate liver expression of hepcidin with the expression of ERFE and TFR2 in murine bone marrow and spleen at 4, 16, 24, 48, 72 and 96 h following administration of a single dose of EPO. Splenic *Fam132b* expression increased 4 h after EPO injection; liver hepcidin mRNA was decreased at 16 h. In the spleen, expression of TFR2 and transferrin receptor (TFR1) proteins increased by an order of magnitude at 48 and 72 h after EPO treatment. The EPO-induced increase in splenic TFR2 and TFR1 was associated with an increase in the number of *Tfr2*- and *Tfr1*-expressing erythroblasts. Plasma exosomes prepared from EPO-treated mice displayed increased amount of TFR1 protein; however, no exosomal TFR2 was detected. Overall, the results confirm the importance of ERFE in stress erythropoiesis, support the role of TFR2 in erythroid cell development, and highlight possible differences in the removal of TFR2 and TFR1 from erythroid cell membranes.

## 1. Introduction

Iron is a central component of hemoglobin. Accordingly, functional erythropoiesis is critically dependent on adequate iron supply, as clearly illustrated by the pathophysiology of iron-deficiency anemia [1]. Developing erythroblast express high levels of transferrin receptor (TFR1) which mediates internalization of circulating diferric transferrin by receptor-mediated endocytosis to ensure adequate iron uptake for hemoglobin synthesis [2].

In addition to TFR1, erythroid cells also express transferrin receptor 2 (TFR2). TFR2 has lower affinity for circulating transferrin [3], and its role in iron acquisition remains unclear. Whereas mice deficient in TFR1 die in utero with severe iron deficiency [4], *Tfr2*-mutated mice display iron overload [5] due to increased iron absorption from the intestine. In contrast to the ubiquitously expressed transferrin receptor, TFR2 is mainly expressed in hepatocytes where it plays an important role in the regulation of iron absorption [5,6]. Although *Tfr2* mRNA is also found in erythroleukemic cell lines and in murine erythroblasts [3], global deletion of the *Tfr2* gene in mice results only in a mild change of hematologic parameters [7,8], strongly suggesting that the primary physiological function of TFR2 is the regulation of dietary iron absorption, rather than iron delivery to erythroid cells. Nevertheless, the potential role of TFR2 in erythropoiesis and its regulation has recently received more attention [9]; at present, the exact role of TFR2 in erythroid cells is still unclear and under active investigation [10,11,12,13,14].

Transferrin-bound iron is regarded as the major physiological source of iron for erythropoiesis, as documented by severe anemia in hypotransferrinemic mice [15]. During states of accelerated erythropoiesis, it is imperative to increase iron export from iron-storing macrophages into plasma in order to ensure adequate iron loading of circulating transferrin. To this purpose, accelerated erythropoiesis downregulates the synthesis of hepcidin, the systemic regulator of cellular iron export. Hepcidin is a small hepatocyte-derived peptide whose primary role is to block iron export from duodenal enterocytes by degradation and inhibition of ferroportin. It has been known for almost twenty years that accelerated erythropoiesis decreases hepcidin (*Hamp*) expression in mice [16,17,18]. The mechanism of this erythropoiesis-mediated decrease of *Hamp* expression was elucidated only in 2014 when it was demonstrated that, upon stimulation by erythropoietin, erythroblasts produce erythroferrone (ERFE), a secreted protein which upon its release into circulation downregulates hepatic hepcidin synthesis [19]. Although it was originally postulated that hepatocytes express a specific receptor or receptors for ERFE [19], it has later been demonstrated that ERFE acts by binding and inactivating BMP6 protein and the BPM6/BMP2 heterodimer [20,21,22]. Bone morphogenetic proteins (BMPs) are major physiological regulators of hepcidin expression [23]. The BMP2 and BMP6 proteins are synthesized and secreted by liver sinusoidal endothelial cells, and, upon interacting with the bone morphogenetic protein receptor complex, stimulate hepcidin synthesis. Inactivation of *Bmp6* or *Bmp2* genes in mice is known to dramatically decrease *Hamp* expression [24,25]; therefore, sequestration of the BMP6 and BMP2 by circulating proteins such as ERFE can be expected to decrease hepcidin synthesis.

Very interestingly, it has recently been reported that TFR2 is able to bind the BMP2 protein and to affect BMP2-dependent signaling during bone development [26]. Theoretically, in analogy with ERFE, increased synthesis of TFR2 could thus contribute to hepcidin downregulation by binding of the BMP proteins. In the case of ERFE, it is known that its basal expression in erythroblasts is very low, but that it is rapidly induced following erythropoietin administration [19], with significant increase in ERFE (*Fam132b*) mRNA detectable already 4 h after EPO injection. No such information is yet available for TFR2, although it has been noted that repeated injections of erythropoietin result in marked increase in both *Tfr2* mRNA and TFR2 protein in the spleen [27]. It is not known whether this increase represents a direct induction of TFR2 expression by EPO, or whether it is caused by EPO-induced expansion of a particular population of erythroid precursors.

The TFR1 and TFR2 proteins are to a significant degree similar. The classical transferrin receptor is expressed at all stages of murine erythroblast development [28], but as the erythroblasts develop into reticulocytes, it is removed from the cell surface and circulates in plasma, where it can be measured for diagnostic purposes [29,30]. The sequence at which the human TFR1 protein is cleaved during receptor shedding is absent from TFR2; nevertheless, shedding of human TFR2 has been described in in vitro experiments [31]. If TFR2 is cleaved from the membrane in vivo, it could circulate in plasma similarly to ERFE; alternatively, plasma TFR2 could be synthesized and secreted as the soluble TFR2 β isoform [8]. In the case of TFR1, the initial removal of the protein from reticulocyte membrane is mediated by exosomes [32], small secretory vesicles which are loaded with plasma membrane proteins and subsequently released from the cell. TFR1 is one of the most abundant proteins found in plasma exosomes; it has not yet been reported whether TFR2 could be removed from erythrocyte precursors by the same mechanism. Overall, compared to TFR1, several aspects of in vivo biosynthesis and processing of TFR2 in erythroid cells are still unclear, including the mechanism of its induction by EPO, its removal from the cell membrane, and its potential presence in plasma.

The purpose of this study was to determine the time course of *Tfr2* mRNA induction in the bone marrow and spleen following a single dose of erythropoietin, and to correlate it with the time course of hepatic hepcidin downregulation. In addition, the content of *Tfr1*, *Tfr2* and *Erfe* mRNA was determined in erythroblast populations obtained by flow cytometry sorting after EPO administration. The results indicate that ERFE and TFR2 display different mechanisms of erythropoietin-mediated induction: Whereas erythroferrone is very rapidly induced in the bone marrow and the spleen by a transcriptional mechanism, the marked increase of TFR2 protein in the spleen following erythropoietin administration is mainly dependent on the increase of the number of TFR2-expressing cells. Comparison of the time courses of erythroferrone and TFR2 induction with the time course of hepcidin downregulation suggests that the role of ERFE in hepcidin suppression is dominant, with marked TFR2 induction occurring only after hepcidin downregulation. As to the mechanism of TFR1 and TFR2 shedding from erythrocyte precursors, plasma exosomes isolated from EPO-treated mice displayed increased amounts of TFR1, whereas no exosomal TFR2 protein could be detected. These results confirm the important role of exosomes in TFR1 processing, and suggest that the removal of TFR2 from the membrane of erythroid cells probably occurs by an exosome-independent mechanism. Most importantly, the similar time course of EPO-induced expression of TFR1 and TFR2 in the spleen supports a role of TFR2 in erythropoiesis regulation.

## 2. Results

### 2.1. EPO Administration Induces Splenic TFR2 Protein Content at 48 and 72 h Post Injection

Administration of a single dose of EPO (200 U/mouse) resulted in the expected increase of red blood cell parameters, which reached statistical significance at 48 h post application and later (Appendix A). With the exception of the 24 h interval, plasma iron content was not significantly changed by EPO treatment (Appendix A). *Hamp* mRNA content started to decrease at 16 h post application, at 24 h the decrease was statistically significant (Figure 1). The decreased expression of hepcidin was associated with a decrease in BMP/SMAD signaling, as evident from the decrease in *Id1* mRNA content (Figure 1). Both bone marrow and spleen *Fam132b* mRNA content were dramatically increased at 4 h after EPO administration (Figure 2a,b). *Hamp* mRNA as well as *Fam132b* mRNA content returned to control values 96 after injection (Figure 1 and Figure 2).

*Tfr2* mRNA in the bone marrow was increased at 4, 16, 48 and 72 h after EPO injection, maximal increase was approximately threefold (Figure 2c). Similar time course and magnitude was observed for marrow *Tfr1* mRNA induction (Figure 2e). In the spleen, no increase in *Tfr2* or *Tfr1* mRNA was observed at 4 and 16 h; however, at 24 h, both *Tfr2* and *Tfr1* mRNA started to increase, with maximum values, exceeding an order of magnitude, observed after 48 and 72 h respectively (Figure 2d,f). Approximately similar time course, with peak values reached after 72 and 48 h post injection, was also observed for erythropoietin receptor (*Epor*) mRNA (Appendix A) and *Cdc42bpa* mRNA encoding the serine/threonine-protein kinase MRCKα (Appendix A), a recently identified TFR2 binding partner [14]. The splenic expression of *Mcoln1* and *Mfn1*, two genes recently reported to be co-expressed together with TFR2 in human erythroblasts [11], was only slightly (less than two-fold) altered by EPO treatment (Appendix A), in contrast with the marked increase in splenic *Tfr2* mRNA (Figure 2d).

The increase in splenic *Fam132b*, *Tfr2* and *Tfr1* mRNA content was closely mirrored by an increase of the respective proteins in splenic membrane fraction obtained by ultracentrifugation (Figure 3).

### 2.2. EPO Administration Increases the Number of Erythroid Precursors in the Spleen

To determine the effect of EPO administration on the number of erythroid precursors in the spleen and bone marrow, cells were sorted at 24, 48 and 96 h post injection according to the Ter119/CD44 pattern [28]. As can be seen in Figure 4, EPO treatment resulted in significant increase of the number of proerythroblasts and basophilic erythroblasts in the spleen at the 48 h interval. Since proerythroblasts are reported to display a higher level of *Tfr2* expression in comparison with other erythroblast fractions [9], the marked increase in splenic TFR2 protein seen at 48 h post EPO injection (Figure 3) can be probably explained by the observed increase in proerythroblast numbers at this time point (Figure 4).

### 2.3. EPO Administration Increases the Content of Fam132b mRNA, but Not Tfr2 or Tfr1 mRNA, in Erythroid Precursors

To examine the effect of EPO treatment on the mRNA content of *Tfr2* and *Tfr1* in erythroid precursors, populations of proerythroblasts, basophilic erythroblasts and polychromatophilic erythroblasts were isolated by FACS sorting [28] from control and EPO treated bone marrow and spleen 48 h after EPO injection. As expected [19], EPO treatment resulted in dramatic upregulation of *Fam132b* mRNA in all three precursor populations. In contrast, *Tfr2* and *Tfr1* mRNA content was not significantly increased (Figure 5). As previously noted [9], the amount of *Tfr2* mRNA was markedly higher in proerythroblasts than in basophilic and polychromatophilic erythroblasts; however, we did not observe the reported [9] correlation between *Tfr2* and *Epor* expression, as, in contrast to *Tfr2* expression, *Epor* expression remained high in both basophilic erythroblast and polychromatophilic erythroblast (Figure 5).

### 2.4. TFR2 Is Not Detected in Mouse Plasma Exosomes after EPO Administration

It is very well established that during red blood cell maturation TFR1 is removed from the reticulocyte membrane and released into plasma [29,30]. In the case of TFR2, cleavage and release of the cleaved receptor has been described in vitro [31]. Since the Abcam ab80194 antibody is raised against the extracellular part of TFR2, it could theoretically detect circulating TFR2 in plasma. However, using albumin-depleted serum from *Tfr2*-/- mice as negative control, no soluble TFR2 could be detected in EPO-treated mice (Appendix A), indicating that, in contrast to TFR1, the amount of plasma TFR2 is too low to be detected by immunoblotting.

Removal of TFR1 from reticulocyte membranes occurs through the secretion of exosomes, from whose surface the extracellular part of TFR1 is subsequently cleaved and shed [33]. Because EPO treatment increases both TFR1 and TFR2 protein synthesis, it was of interest to determine whether EPO administration will also increase the TFR1 and TFR2 protein content in plasma exosomes. As shown in Figure 6, exosomes isolated by ultracentrifugation from plasma 72 h after EPO administration displayed increased TFR1-related signal compared to control exosomes. In contrast, no TFR2-related signal could be detected in exosomes from control or EPO-treated mice. These results suggest that the process of receptor removal from the membrane of erythroid cells might be different for TFR1 and TFR2.

## 3. Discussion

The crucial role of TFR1 in providing iron for erythropoiesis is established without any doubt [2]. In contrast, the exact function of TFR2 in erythroid cells is still unclear. Recently published data suggest that TFR2 could theoretically participate in bone morphogenetic protein binding [26]—a mechanism which closely resembles the recently proposed mode of action of ERFE [20,21,22]. Whereas the rapid induction of ERFE by EPO is well documented, much less is known about the time course of EPO-induced synthesis of TFR2. Therefore, the primary purpose of this study was to compare the expression of TFR2 with the expression of ERFE and TFR1 following a single dose of EPO. The results show a marked difference in the kinetics and mechanism of EPO-mediated induction between ERFE and both transferrin receptors.

ERFE has been originally discovered as a protein which mediates hepcidin downregulation during stress erythropoiesis [19]; accordingly, its expression rapidly increases following EPO injection. In contrast, the effect of EPO on the synthesis of *Tfr2* mRNA in bone marrow is much less pronounced. However, an intriguing pattern of *Tfr2* mRNA induction was observed in the spleen, where significant increase of *Tfr2* mRNA occurs at 48 and 72 h after EPO treatment (Figure 2d). Importantly, immunoblot analysis demonstrated that the observed increase in *Tfr2* mRNA is closely mirrored by an increase in the content of splenic TFR2 protein. Since spleen is the main site of EPO-induced stress erythropoiesis [34], the observed marked increase of splenic TFR2 protein suggests a possible role of TFR2 in mouse stress erythropoiesis.

The mechanism underlying the increase of TFR2 protein content in the spleen at 48 h and 72 h after EPO injection could theoretically involve direct EPO-mediated transcriptional increase of *Tfr2* mRNA in individual cells, as is the case with ERFE [19], or it could reflect the EPO-induced increase of the number of TFR2-producing cells. In contrast to *Fam132b* mRNA, which was in accordance with previous studies [19] robustly induced in splenic proerythroblasts and basophilic erythroblasts, the effect of EPO on *Tfr2* mRNA content in individual erythroid cell populations was not statistically significant (Figure 5). On the other hand, EPO significantly increased the number of proerythroblasts and basophilic erythroblasts in the spleen 48 h after treatment (Figure 4). Therefore, we conclude that the substantial increase in TFR2 protein observed in the spleen at 48 h and 72 h after EPO administration is caused by an increase in the number of TFR2-expressing cells, whereas the increase in ERFE protein reflects both the increase in ERFE-producing cells, as well as increased transcription of the *Fam132b* gene in individual cells. In accordance with previously published data [9], *Tfr2* expression was markedly higher in proerythroblasts than in the later precursor populations; in contrast, *Epor* expression remained relatively high even in the late precursors (Figure 5) indicating that, in mice, the two genes are differently regulated during the later stages of erythroblast development.

Based on the recently reported TFR2/BMP2 interaction [26], the increased synthesis of TFR2 following EPO administration could theoretically contribute to the downregulation of hepcidin. However, comparison of the time course of hepcidin downregulation with the induction of ERFE and TFR2 proteins points to a major role of ERFE, rather than to a role of TFR2. In agreement with previously published data [19,35,36], *Hamp* expression decreases in less than 16 h following a single injection of EPO, whereas, as demonstrated in the present study, increased splenic TFR2 protein synthesis is seen only at 24 h and later. It is therefore evident that the early decrease in hepcidin expression must be related primarily to the induction od ERFE synthesis, or to the decrease of monoferric transferrin [35]. Nevertheless, since it has been reported that prolonged administration of EPO can downregulate hepcidin expression even in *Fam132b*-/- mice [37], the contribution of BMP-binding molecules other than ERFE to hepcidin downregulation can not be ruled out. In this respect, soluble TFR2 [31] or the beta isoform of TFR2 [8] represent interesting potential candidates.

In contrast to TFR2, whose role in erythroid cells is not yet completely clear, the function of TFR1 in erythropoiesis is established without any doubt [2]. Transferrin iron is regarded as the main physiological source of iron for hemoglobin synthesis, and TFR1 molecules are indispensable for efficient hemoglobinization. It is therefore interesting to note that the time course of splenic TFR1 and TFR2 induction following EPO administration is similar for both receptors, despite the known fact that the expression of both genes is regulated by different mechanisms. *Tfr1* mRNA has five iron-responsive elements in its 3-untranslated region [38], which mediate the induction of TFR1 during states of iron deficiency; no such sequences are present in *Tfr2* mRNA. Regardless, the time-response data following EPO administration are roughly similar for both receptors (Figure 2 and Figure 3), supporting the concept that, in erythroid cells, the iron-responsive element-dependent control of *Tfr1* expression is overridden [39] and erythroblast continue to take up iron despite high iron loading.

The approximately similar time course, as well as the approximately similar magnitude of splenic TFR1 and TFR2 induction following EPO treatment, indirectly support the hypothesis that TFR2 has an important role in erythroid development [9,10]. One such role could be iron acquisition from lysosomes [11]. Based on a similar expression pattern of *TFR2*, *MCOLN1* and *MFN2* genes, it has been suggested that TFR2 protein delivers iron from lysosomes to mitochondria, in cooperation with the mucolipin-1 and mitofusin-2 proteins [11]. However, the present study did not find significant correlation between splenic *Tfr2* and *Mcoln1* or *Mfn2* mRNA, and thus does not provide additional experimental support for this proposed role of TFR2 in iron homeostasis.

The TFR1 and TFR2 proteins are to a considerable extent similar, with analogous internalization motifs, domain organizations and disulfide bridges; both proteins exist as homodimers and it has even been reported that they can form heterodimers [40]. One substantial difference between them is the lack, in the TFR2 protein, of a cleavage sequence around arginine 100 at which human TFR1 is cleaved from the membrane by the protease PC7 [41]. Despite this discrepancy, the release of soluble TFR2 from an erythroleukemic cell line has been reported in in vitro experiments [31]. The removal of TFR1 from reticulocyte membranes is mediated by exosomes [32,33]; accordingly, plasma exosomes from EPO-treated mice were found to display increased amount of full-length TFR1 protein (Figure 6). At present, there is little information on the mechanism of removal of TFR2 from the membrane of erythroid cells. Due to the similarities between the two receptors, it was expected that plasma exosomes would contain, in addition to TFR1, detectable amount of TFR2. However, no TFR2-related signal was observed in plasma exosomes from control or EPO-treated mice. This failure to detect exosomal TFR2 could reflect the decrease of TFR2 protein content during later stages of erythroblast maturation, which has recently been reported in several cellular models of murine terminal erythroid differentiation [42], or it could be simply caused by low sensitivity of the antibody used. On the other hand, it could also point to a difference in the mechanism of removal of the two receptors from the membrane of murine erythroid cells in vivo. Since exosomal release of TFR2 protein has been reported in K562 cells [43,44], the role of exosomes in TFR2 processing remains a potential subject for further studies.

In conclusion, the presented study reports, at both mRNA and protein level, robust induction of TFR2 protein in the spleen at 48 and 72 h after a single injection of EPO. The underlying mechanism is probably the EPO-induced increase in TFR2-expressing cells in the spleen. The time course and magnitude of induction was approximately similar for both TFR2 and TFR1, despite the known absence of an IRE-IRP system in TFR2. Comparison of the time course of *Hamp* mRNA downregulation with the induction of *Fam132b* and *Tfr2* mRNA suggests that the effect on hepcidin synthesis in this experimental setting is mediated primarily by ERFE; nevertheless, contribution of TFR2 to hepcidin downregulation following prolonged administration of EPO can not be ruled out. Finally, the absence of detectable TFR2 in plasma exosomes suggests that each of the two transferrin receptors has a different mechanism for its removal from the cell membrane.

## 4. Materials and Methods

### 4.1. Animals and Treatment

All experiments were approved by the Ministry of Education of Czech Republic, protocol MSMT-11192/2020-2, dated April 28, 2020. Male C57BL/6 mice aged 8 weeks were treated by a single intraperitoneal injection of EPO (NeoRecormon Roche 2000, Roche Diagnostics GmbH, Penzberg, Germany) diluted in PBS at 200 U/mouse, control mice received PBS only. Animals were euthanized by ether anesthesia. Heparinized blood was obtained by retrobulbar puncture in ether anesthesia, red blood cells parameters were determined on Advia hematologic analyzer. Hematocrit values were determined by centrifugation in microtubes. Plasma iron was measured by a commercial kit (Fe Liquid, Erba-Lachema s.r.o., Brno, Czech Republic).

### 4.2. RNA Analyses

Liver and spleen tissue was stored in RNA Later; RNA was extracted using Qiagen RNEasy Plus Mini kit. For bone marrow RNA extraction, bone marrow from femurs was flushed directly into the Qiagen RLT Plus Buffer containing mercaptoethanol. Reverse transcription was performed by RevertAid kit (Thermo Fisher Scientific, Waltham, MA, USA). Real-time PCR analysis was run using BioRad SYBR Green Mix (Bio-Rad, Hercules, CA, USA) on a BioRad IQ5 cycler; beta actin (*Actb*) was used as a reference gene. Primer sequences are given in Appendix A.

### 4.3. Immunoblotting

Spleen ERFE, TFR2 and TFR1 proteins were determined in cell membranes obtained by ultracentrifugation. Spleen samples (40–60 mg) were homogenized in 1 mL of 10 mM Hepes buffer, pH 7.4, containing 250 mM sucrose and protease inhibitor (Complete Mini, Roche, Sigma-Aldrich, Prague, Czech Republic). Homogenization was performed by 6 mm Ultra Turax homogenizer (3 × 10 s at maximum speed). After 30 min on ice, homogenates were centrifuged for 15 min at 8000× *g* and the supernatant was subjected to ultracentrifugation at 100,000× *g* for one hour. Pellets were resuspended in 500 µL of homogenization buffer and washed by another ultracentrifugation round; subsequently, washed pellets were resuspended in 75 µL of 25 mM ammonium bicarbonate containing 2% SDS. Proteins were separated under reducing conditions on 8% polyacrylamide gels, samples were heated at 85 °C for 10 min prior to loading. Immunoblotting was performed on Invitrogen SureLock blotter using PVDF membranes. For the detection of full-length TFR2 protein in the spleen, the TFR21-A antibody from Alpha Diagnostics Intl. (San Antonio, TX, USA) raised against the intracellular part of TFR2 was used. Exosomes and serum were probed for TFR2 presence using Abcam ab80194 antibody raised against the extracellular part (AA 150–250) of human TFR2. The specificity of this antibody was verified on liver samples from *Tfr2*-/- mice (Appendix A). Spleen TFR1 was detected using Abcam ab61134 antibody; exosome and serum TFR1 was detected by Abcam ab214039 antibody.

### 4.4. Flow Cytometry and Cell Sorting

Bone marrow cells were obtained from the long bones (femurs and tibias) by flushing the bone cavity with PBS supplemented with 1% bovine serum albumin through a hole in one end of the bone without clipping off the epiphyses. Spleen tissue was disrupted by a loosely fitting glass homogenizer. Single-cell suspension was obtained by repeated passage through a 25G needle and filtered through a 70 μm nylon cell filter. Populations of erythrocyte precursors were identified by flow cytometry as based on the Ter119/CD44 pattern [28]. Gating strategy and experimental details are described in Appendix A. Cells were stained by fluorescently labeled antibodies for 20 min at 4 °C in the dark with antibodies listed in Appendix A. RNA was isolated from the sorted populations using Qiagen RNEasy Micro kit (Qiagen N.V., Venlo, The Netherlands); 50,000 cells were sorted directly into 350 µL of RLT buffer without mercaptoethanol, homogenized by vortexing for 60 s and after adding 375 µL of 70% ethanol, processed according to the manufacturer instruction.

### 4.5. Exosome Preparation

Plasma exosomes were obtained by ultracentrifugation [32] from 1.5 mL of heparinized mouse plasma, the pellet was washed once with PBS. The final exosome pellet was resuspended in 30 µL of 2% SDS in 25 mM ammonium bicarbonate. Average yield of exosome protein was approximately 5 μg of protein per ml of plasma.

### 4.6. Statistical Analysis

Values are graphed as mean ± SD. Values between control groups and EPO-treated groups were compared by unpaired Students *t*-test. *p* ˂ 0.05 is regarded as significant.

## Figures and Tables

**Figure 1 ijms-22-08209-f001:**
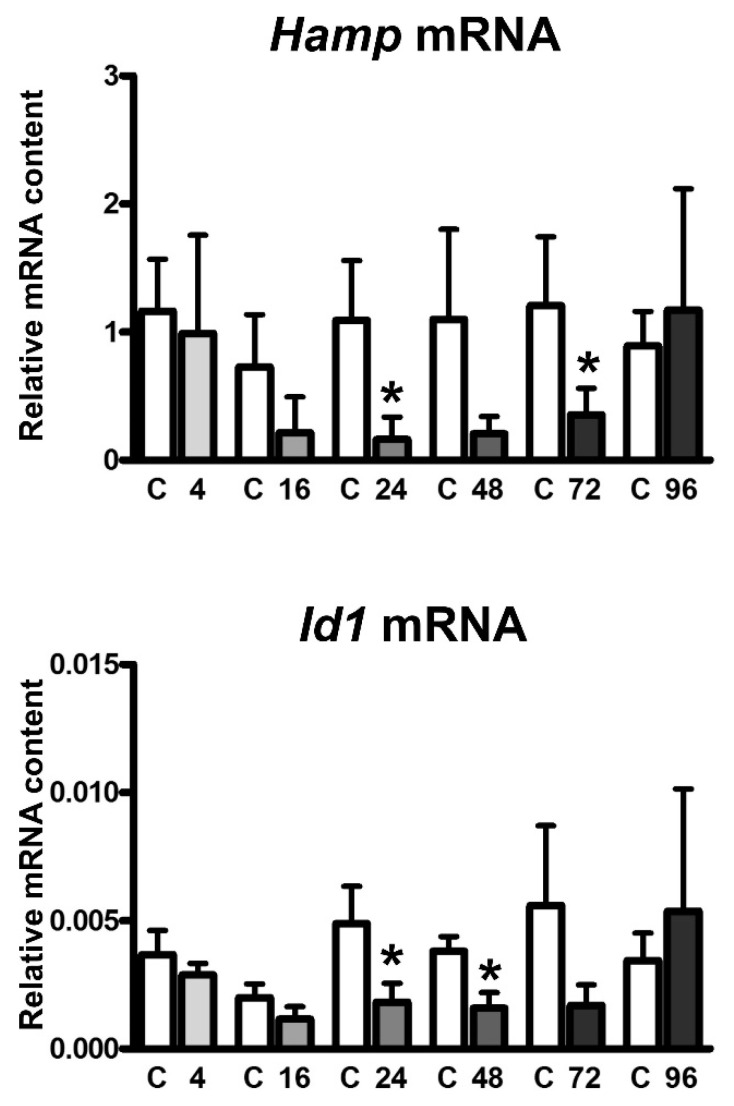
Single injection of EPO downregulates hepcidin expression at 16 h post injection. EPO was administered at 200 U/mouse and the relative expression of liver *Hamp* and *Id1* was determined at the indicated time points in groups of three control mice (C) and three EPO-treated mice. Target mRNA content is expressed relative to *Actb* expression, * denotes statistical significance between EPO-treated group and the respective control group (*p* < 0.05). Values were obtained from two independent experiments.

**Figure 2 ijms-22-08209-f002:**
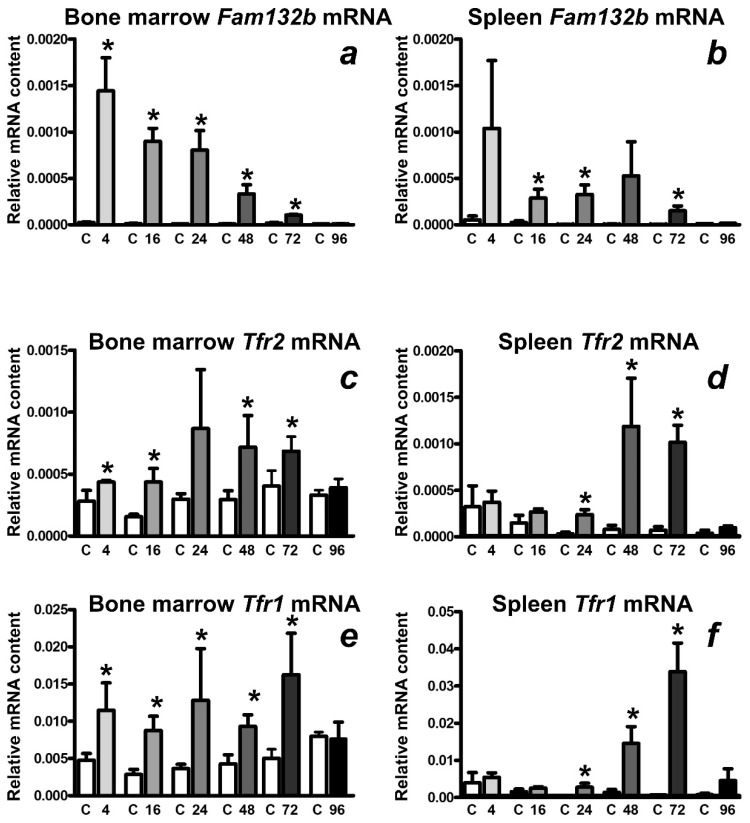
(**a**,**b**) Single injection of EPO rapidly upregulates *Fam132b* mRNA in the bone marrow and spleen. (**c**,**d**) Single injection of EPO upregulates splenic *Tfr2* mRNA at 48 h and 72 h after injection. (**e**,**f**) Single injection of EPO upregulates splenic *Tfr1* mRNA at 48 h and 72 h after injection. EPO was administered at 200 U/mouse, mRNA content was determined at the indicated time points in groups of three control mice (C) and three EPO-treated mice. Target mRNA content is expressed relative to *Actb* expression, * denotes statistical significance between EPO-treated group and the respective control group (*p* < 0.05). Values were obtained from two independent experiments.

**Figure 3 ijms-22-08209-f003:**
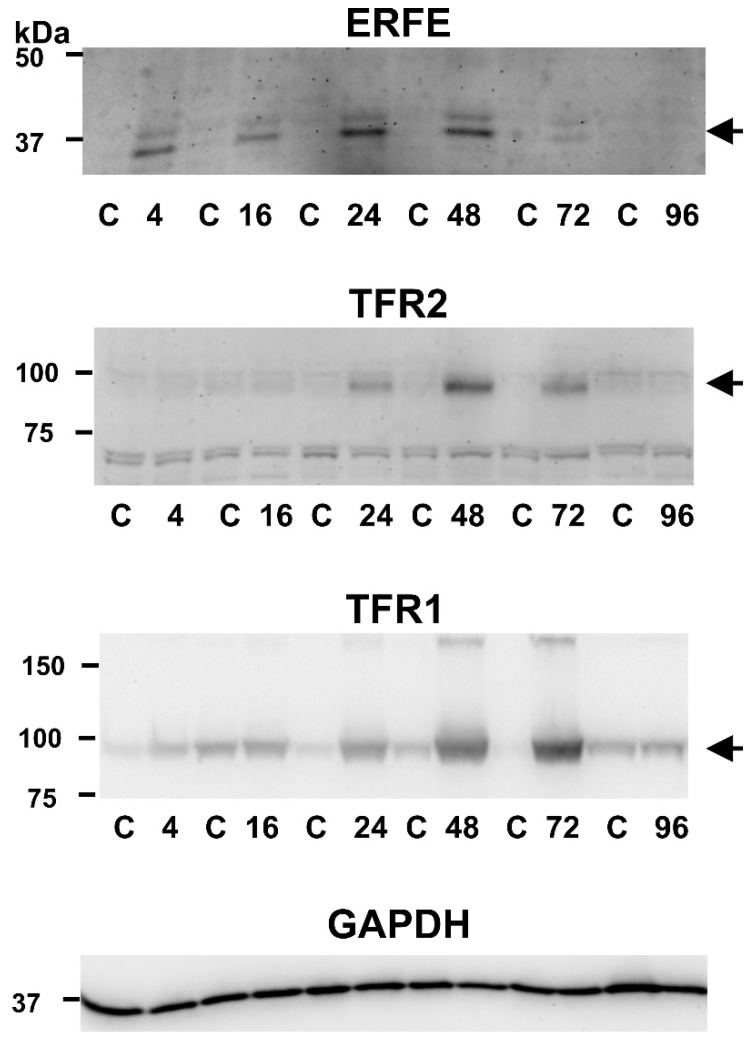
Single injection of EPO upregulates ERFE, TFR2 and TFR1 protein in the spleen. EPO was administered at 200 U/mouse; target proteins were determined at the indicated time points (4, 16, 24, 48, 72 and 96 h after EPO administration) in spleen membranes isolated from pairs of control (C) and EPO-treated animals. Sample loading is 35 µg protein/well, GAPDH is used as loading control. Arrows denote the target protein bands.

**Figure 4 ijms-22-08209-f004:**
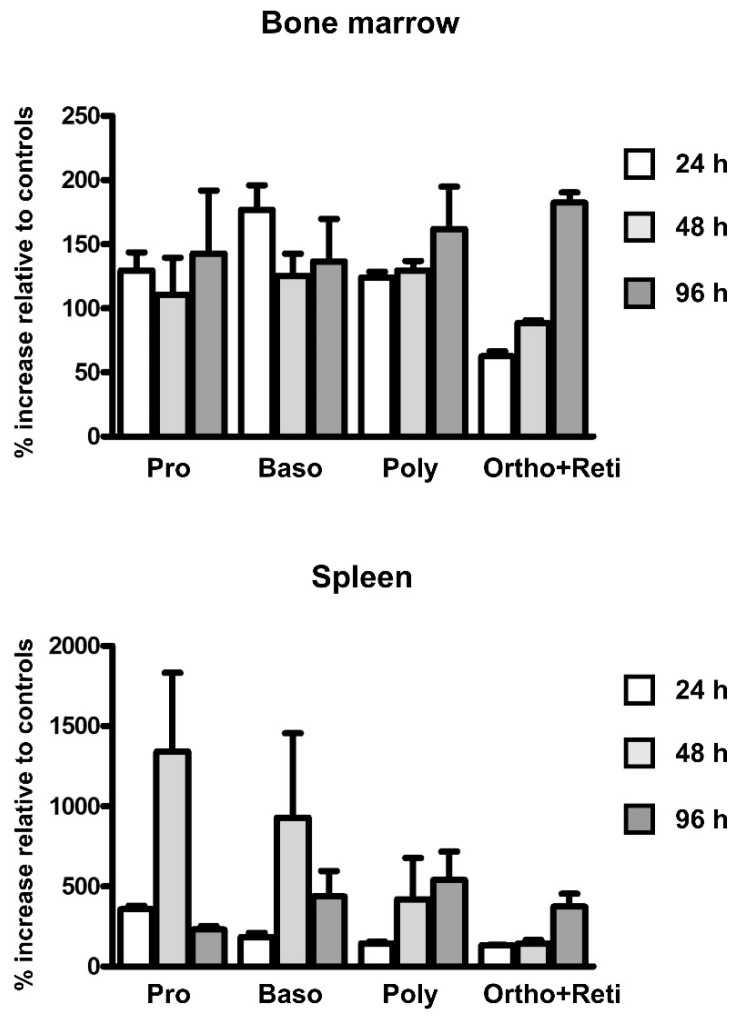
Single injection of EPO increases the number of proerythroblasts and basophilic erythroblasts in the spleen 48 h after treatment. EPO was administered at 200 U/mouse; cell populations were obtained at indicated intervals by FACS sorting. Values represent the relative increase of cell populations in EPO-treated animals as compared to control animals (100%). *n* = 3, values were obtained from three independent experiments. Pro: Proerythroblasts; Baso: Basophilic erythroblasts; Poly: Polychromatophilic erythroblasts; Ortho + Reti: Orthochromatic erythroblasts and reticulocytes.

**Figure 5 ijms-22-08209-f005:**
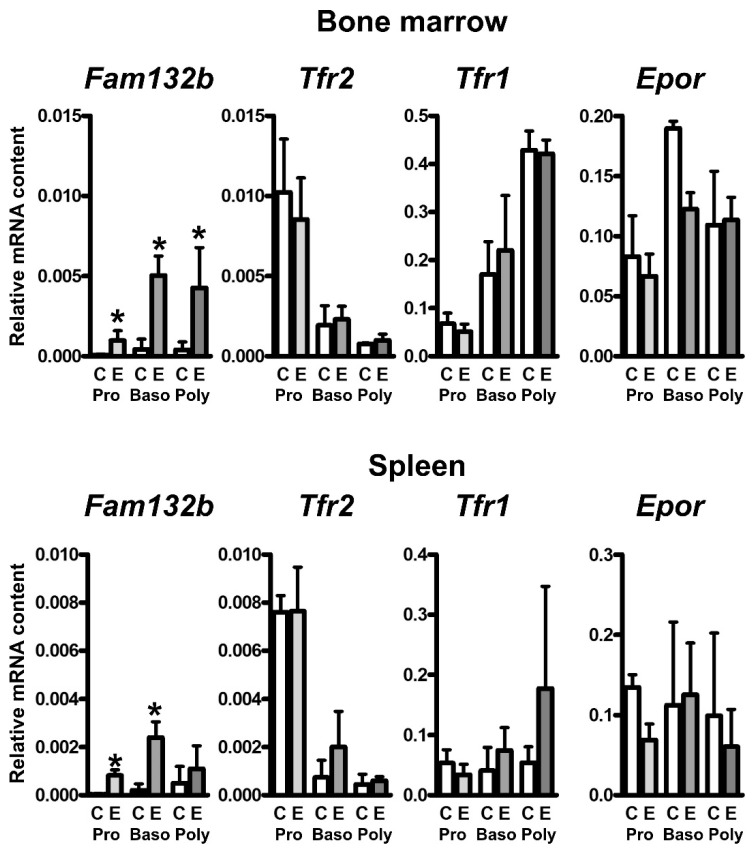
Single injection of EPO increases *Fam132b* expression in bone marrow and spleen erythroid precursors. EPO was administered at 200 U/mouse; cell populations were obtained 48 h after treatment by FACS sorting. C: Expression in control populations, E: expression in EPO-treated populations. Pro: Proerythroblasts; Baso: Basophilic erythroblasts; Poly: Polychromatophilic erythroblasts. Target mRNA content is expressed relative to *Actb* expression, * denotes statistical significance between EPO-treated group and the respective control group (*p* < 0.05). *n* = 3, values were obtained from three independent experiments.

**Figure 6 ijms-22-08209-f006:**
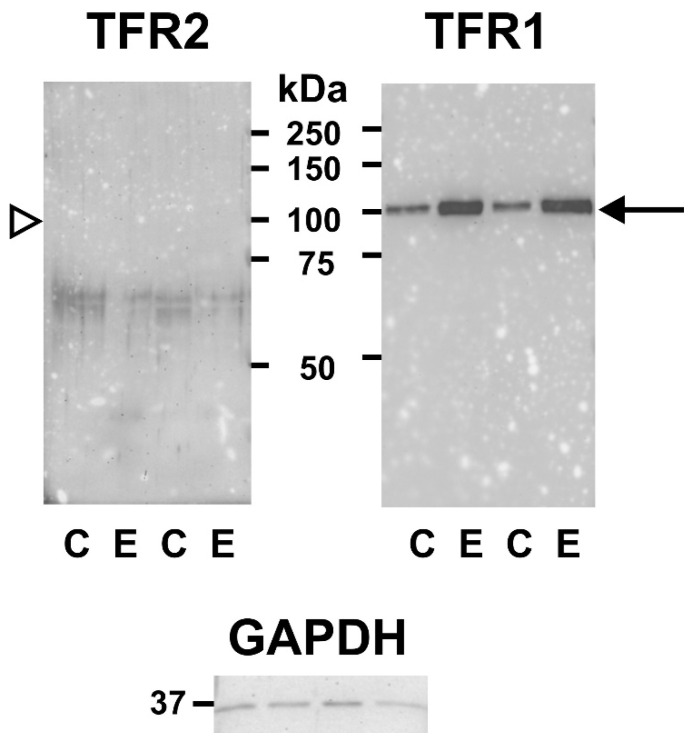
Single injection of EPO increases TFR1 protein content in plasma exosomes. Immunoblotting for TFR2 and TFR1 in plasma exosomes isolated from control (C) and EPO (E) treated mice. EPO was administered at 200 U/mouse; mice were sacrificed 72 h after EPO injection. Arrow denotes the TFR1-specific band, arrowhead indicates the expected size of exosomal TFR2, which was not detected. The faint bands around 65 kDa in the TFR2 panel are probably nonspecific. GAPDH is used as loading control.

## Data Availability

The data presented in this study are available in the article and Appendix A. Further details, if needed, are available on request from the corresponding author.

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
