# Peer review of "Effect of Erythropoietin on the Expression of Murine Transferrin Receptor 2"

_ijms, 2021, doi:10.3390/ijms22158209_

Round 1

Reviewer 1 Report

The presented study aims to increase our understanding of the role of Trf2 in the interplay between iron and erythropoiesis.

Whereas the questions raised in the introduction of the manuscript are relevant, the presentation and interpretation of the data are not clear, and sometimes contradictory (e.g.  the effect of EPO on Tfr2 mRNA expression, line 244-245, line 176-177).

The manuscript is very long (intro and discussion!) and not structured logically, making it difficult to read and impossible to draw conclusions based on descriptive data and associations.

The manuscript can be improved as a letter, using the most important data on Trf2 (and TRFC) expression directly related to EPO injection and hepcidin and ERFE expression. This can be descriptive, suggesting potential functional consequences following EPO injection with respect to erythropoietic output and iron utilization. In its current form different topics (effect on hepcidin and BMP signaling, shedding or membranes) are distracting from the main subject. 

I miss appropriate loading controls for the western blots (Figure 3, excl ERFE). 

Author Response

We thank the Reviewer for the comments. We agree that various aspects of the manuscript, such as the effect of TFR2 on hepcidin and BMP signaling or the shedding of TFR2 from membranes, are not of major interest to researchers investigating the regulation of erythropoiesis. We have therefore asked for the transfer of the manuscript to the newly available special issue "New Advances in Iron Metabolism, Ferritin and Hepcidin Research", for which the manuscript topics are more appropriate.

In order to improve the structure of the manuscript, we have left out the discussion regarding ferroportin and the corresponding text and Supplemental Figure.

Reviewer 2 Report

The data presented in this article by Berezovsky et al. are initially based on previous observations by the team demonstrating that injections of erythropoietin result in increase in both Tfr2 mRNA and Tfr2 protein in the spleen of mice (Frydlova et al. Plos One 2019). The aim of this article is to determine whether this upregulation is a direct induction of Tfr2 expression in respons to Epo-mediated intracellular signaling or wether it is due to an increase in the number of erythroid progenitors or precursors expressing Tfr2. The study aims to determine the time course of Erfe, Tfr2 and Hepcidin expression in respons to Epo injection in mice. The last question is to establish whether Tfr2 removal from reticulocytes is an exosome-dependent mechanism.

The questions asked are interesting and will help to better understand the role of Tfr2 in erythropoiesis. The manuscript is easy to understand and well written.

As previously demonstrated in the literature, Epo induces a decrease of hepatic hepcidin mRNA expression after 16h of injection (Fig1). The authors show that Epo induces an increase of Erfe mRNA expression in the bone marrow (BM) and the spleen after 4h of injection. Tfr2 mRNA level is increased after 16h in the BM and 24h in the spleen, for Tfr1 the kinetics is 4h in the BM and 24h in the spleen (Fig2). These results have been confirmed in the spleen at the protein level by Western Blots (Fig 3). To determine the origin of the enhanced expression of Tfr2 after Epo injection, the authors sort by Facs the different erythroid precursors (Fig 4) and show that Epo induced a dramatic increase of Fam132b mRNA as previously demonstrated in the literature. However, the mRNA content of Tfr2, Tfrc and Epor is stable demonstrating that Epo enhance the number of Tfr2 expressing cells but not the level of mRNA in each erythroid precursor (Fig5).

The data show that Epo injection induces an increase in TFRC-expressing exosomes but no signal was detected by anti-TFR2 WB (Fig6). The author conclude that plasma exosomes do not contain TFR2 protein.  

The authors need to address some issues related to the manuscript:

One major issue is raised:

The authors conclude that plasma exosomes do not contain TFR2 protein and suggest that the removal of TFR2 from reticulocyte membrane is done by an exosome-independent mechanism, contrary to TFR1 removal. To address this question, Tfr2 protein expression in orthochromatic erythroblasts and/or young reticulocytes has to be shown. Indeed, an alternate explanation is the very low quantity of TFR2 compared to Tfr1 in murine erythroblasts. Global proteomic analysis has shown the overall expression of murine Tfr2 is 103 time lower than the one of TfR1. Murine Tfr2 protein expression is maximal at the progenitor stages and then decrease along erythropoiesis to become undetectable by mass spectrometry at the protein level at the end of terminal erythropoiesis (Gauthier et al. Blood Advance 2020).  

 Few minor issues are raised:

- qPCR are performed with only one gene reference, two or three genes reference would reinforce the data.

-Line 86, is the beta- isoform of Tfr2 a secreted or an intracellular protein?

-Line 184: “Tfr2 mRNA was markedly higher” IN “proerythroblasts”, in is missing.

-For statistics, t-test was used, did the authors use unpaired t-test which is the appropriated test? The number of mice or the number of independent experiments has to be indicated for each figure.

Author Response

We thank the Reviewer for the very helpful comments. We were not aware of the TFR2 data included in the Gautier et al. 2020 Blood Advances paper, which to a significant degree affect the discussion of our results.

Gautier et al. report the absence of TFR2 protein from later-stage erythroblasts and reticulocytes. Based on their data, we checked Tfr2 mRNA in FACS-sorted orthochromatic erythroblasts and reticulocytes; our preliminary results confirm the absence of Tfr2 mRNA in these populations. We therefore changed our manuscript accordingly, and instead of reticulocyte TFR2, we now more broadly discuss “erythroid cell TFR2”. We also found one reference to TFR2 protein presence in K562-derived exosomes (Sadvakassova et al, reference 44). Given the marked increase of TFR2 protein in spleens of EPO-treated mice, we believe that a search for the possible presence of TFR2 in plasma exosomes still represents a valid issue for futher experiments, but we completely agree that exosomal TFR2 (if it exists) is very probably not derived from reticulocytes.   

Minor issues:

We re-checked our major results with Rpl13a as a reference gene – while the threshold cycle values for Rpl13a were generally higher (by two to three cycles) than those for beta actin, and therefore resulted in higher relative abundance of the mRNAs investigated, the use of Rpl13a certainly did not affect the observed expression patterns and conclusions.   

According to Roetto et al., Blood 115 (2010) pp 3382-3389, TFR2 beta is an intracellular/secreted protein (p.3383). Although we did not find direct evidence for the beta form of TFR2, we discuss it as a theoretical possibility.

Line 184 has been corrected.

Unpaired t test was used for statistics, this is now specified in Materials and Methods. The number of samples and experiments has been added to figure legends.

Once again, we thank the Reviewer for the important major issue comment.

Reviewer 3 Report

Typing errors correct only 

Author Response

We thank the Reviewer for the evaluation of the manuscript. We took care to correct all typing erors identified by the Word spelling check feature.

Reviewer 4 Report

Interesting and scientific work. There are some things to correct and add in the figures and tables. In all the figures and also in the table, the p of significance is missing. In table 1S it is said that iron does not change (33 +/- 7 is not significant?). I would have put a few words on iron in the discussion. Line 132 is missing the 4-hour increase represented by fig 2a (marked as significant). Figure 2 is not well explained. In the results we speak of hamp and then of fam132b which in figure 2 is represented at the bottom. I would divide Figure 2 by first placing the two Fam132b graphs in the bone marrow and spleen as a and b. Then below, separated by a line, the other graphs representing Tfr2 and Tfrc as fig 2 c and d. Also indicate here the p as in all the figures bearing the asterisk. In my opinion there are too many supplement figure.

Author Response

We thank the Reviewer for the comments.

P values were added to figure legends.

The difference in plasma iron between control and EPO 24h group is indeed significant, we apologize for not noticing. We did not discuss the changes in plasma iron, because we feel that the observed changes are too slight, and rather difficult to explain. In our previous experiments, in which EPO was administered over four days, we often observed a decrease in plasma iron coupled with an increase of hematocrit. In our experience (using the available kit and instrumentation), the determination of plasma iron is influenced by several difficult-to-control factors, such as possible hemolysis and spectrophotometer baseline fluctuations, and valid discussion of the contribution of plasma iron to hepcidin regulation would require more samples.

The text describing Figure 2 has been corrected. According to the Reviewer suggestion, the layout of Figure 2 was changed as required and p values were added.

To reduce the amount of supplementary material, we excluded one Figure (ferroportin protein) from the Supplementary Figures.    

Round 2

Reviewer 1 Report

The manuscript has been improved, yet not much has been done with my comments concerning the lenghty introduction and discussion in relation to the amount of data and scientifc interest. This cannot be solved by moving the manuscript to another issue or section 

I am still convinced that a letter format would be better to present these descriptive data and have a more focused message.    

However, based on the other reviews, I stand alone in this, so I will leave this for the editors to decide on. 

Author Response

We thank the Reviewer for his opinion. We agree that the main finding, i.e. the time-course of splenic TFR2 induction, could be best presented as a letter. We, however, believe that the additional information contained in the manuscript could also be of interest to the iron-researching community. Particularly, we would like to present the data on TFR1 in exosomes. Removal of reticulocyte TFR1 in exosomes was first described in the late eighties, but this fact is only very rarely mentioned in iron-related literature. We therefore hope that, by including the exosome data, the manuscript could be useful even in the presented, extended form.

Once again, we thank the Reviewer for his expert opinion.

Reviewer 2 Report

My concerns have been adequately adressed.

Author Response

We would like to thank the Reviewer for his expert opinion - particularly for pointing out the information regarding the absence of TFR2 protein in reticulocytes, which has completely escaped our attention.